# Prokaryotic nanocompartments form synthetic organelles in a eukaryote

Yu Heng Lau [1,2,3], Tobias W. Giessen[1,2], Wiggert J. Altenburg[1,2] & Pamela A. Silver[1,2]

Compartmentalization of proteins into organelles is a promising strategy for enhancing the productivity of engineered eukaryotic organisms. However, approaches that co-opt endogenous organelles may be limited by the potential for unwanted crosstalk and disruption of native metabolic functions. Here, we present the construction of synthetic non-endogenous organelles in the eukaryotic yeast *Saccharomyces cerevisiae*, based on the prokaryotic family of self-assembling proteins known as encapsulins. We establish that encapsulins self-assemble to form nanoscale compartments in yeast, and that heterologous proteins can be selectively targeted for compartmentalization. Housing destabilized proteins within encapsulin compartments afford protection against proteolytic degradation in vivo, while the interaction between split protein components is enhanced upon co-localization within the compartment interior. Furthermore, encapsulin compartments can support enzymatic catalysis, with substrate turnover observed for an encapsulated yeast enzyme. Encapsulin compartments therefore represent a modular platform, orthogonal to existing organelles, for programming synthetic compartmentalization in eukaryotes.

[1] Wyss Institute for Biologically Inspired Engineering, Harvard University, 3 Blackfan Circle, Boston, MA 02115, USA. [2] Department of Systems Biology, Harvard Medical School, 200 Longwood Avenue, Boston, MA 02115, USA. [3] School of Chemistry, The University of Sydney, Eastern AvenueNSW 2006 Sydney, Australia. These authors contributed equally: Yu Heng Lau, Tobias W. Giessen. Correspondence and requests for materials should be addressed to P.A.S. (email: pamela_silver@hms.harvard.edu)

ntracellular compartmentalization is a fundamental strategy used by all organisms to organize and optimize their metabolism. Examples of compartments in nature range from eukaryotic lipid-bound organelles to prokaryotic protein-based containers[1–4], and their functions include sequestering toxic metabolic products, generating distinct biochemical environments, and stabilizing otherwise unstable proteins and biosynthetic intermediates. The ability to incorporate similar functional properties in engineered organisms could lead to significant improvements in metabolic engineering and recombinant protein expression[5,6]. However, efforts to reprogram naturally occurring compartments for synthetic applications are challenging due to their inherent complexity and the large number of different biomacromolecules involved[7–10]. We therefore identified the encapsulin family of self-assembling prokaryotic proteins as a highly engineerable candidate suitable for designing programmable synthetic organelles in eukaryotes[11–14].

Encapsulins are 25–40 nm diameter hollow compartments comprised of 60 or 180 copies of a single self-assembling capsid protein[11,12]. The varied native functions of encapsulins all involve packaging proteins within their interior as part of the self-assembly process to tailor the activity of packaged components. In vivo protein encapsulation is guided by short targeting peptides (TPs), which are located at the *C*-termini of cargo proteins. A large variety of native cargo proteins has been identified in bacteria and archaea, including peroxidases and ferritin-like proteins involved in stress response pathways[11,14–16]. Encapsulins have been the subject of several recent engineering and characterization studies[17–20], and using *Escherichia coli* as a host, it has been shown that packaging of nonnative proteins into the encapsulins from *Thermotoga maritima* and *Brevibacterium linens* can be achieved by fusion of TPs to the intended cargo[21,22].

Given their modularity and programmability, encapsulins are an ideal platform for building synthetic compartmentalization in eukaryotes. In contrast to approaches that leverage existing organelles[23–26], encapsulins have the advantage of being completely orthogonal to endogenous eukaryotic compartments. There is also ample choice of different encapsulin protein variants derived from different families of bacteria and archaea. In particular, the encapsulin from *Myxococcus xanthus* has been structurally characterized, and has the ability to simultaneously package three different proteins in its native form[14].

Here, we present the construction of synthetic organelles in the yeast *Saccharomyces cerevisiae*, based on the *M. xanthus* encapsulin[14]. We show that encapsulin compartments can stabilize heterologous cargo proteins against degradation, co-localize proteins within their interior, and act as nanoreactors for housing enzymatic catalysts (Fig. 1a). In doing so, we demonstrate that protein-based compartments can mimic the ability of eukaryotic organelles to control protein localization and activity in living cells.

## Results

**Self-assembly of cargo-loaded encapsulins in yeast.** Expression and self-assembly of encapsulin compartments in yeast were achieved using a plasmid containing the encapsulin gene *EncA* from *M. xanthus*[14] under the control of the inducible GAL1 promoter (Fig. 1b). A clear induction band was observed by SDS-PAGE for cultures grown in galactose induction media, corresponding to the 32.5 kDa encapsulin monomer (Supplementary Fig. 3). The identity of the band was confirmed to be *EncA* by mass spectrometry (Supplementary Fig. 4). Isolation of the encapsulin compartments was achieved by PEG precipitation

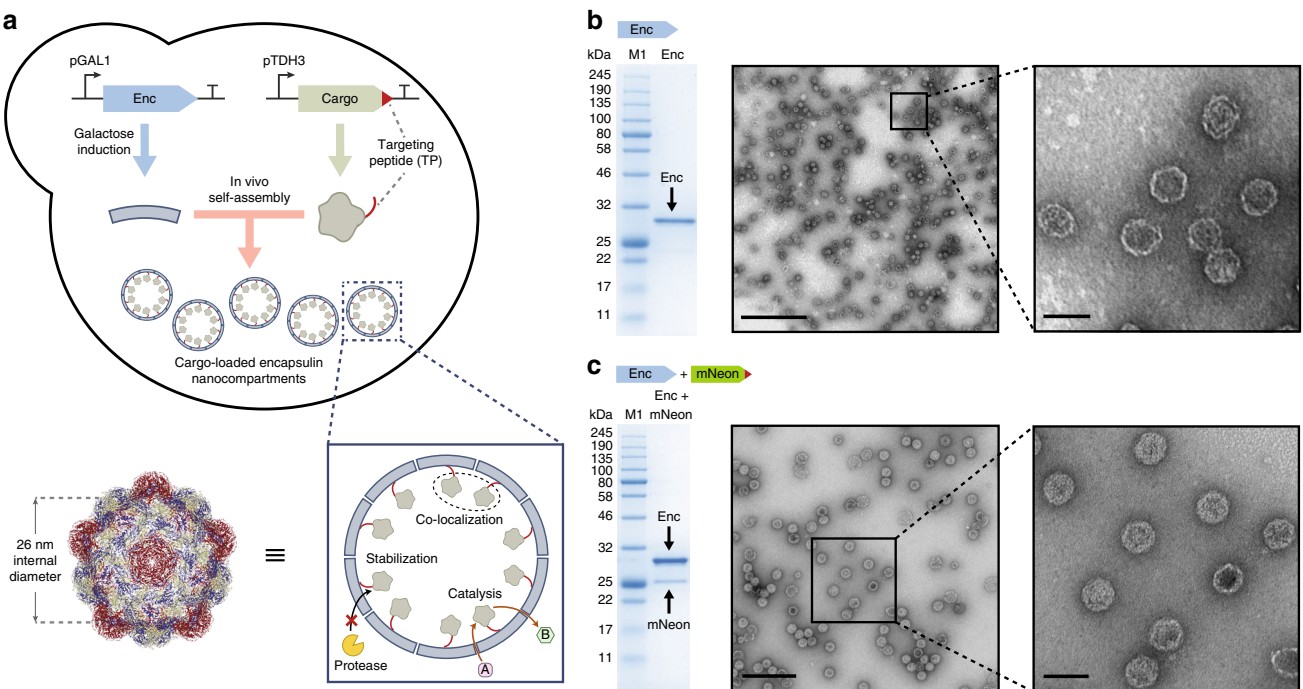

**Fig. 1** Expression, assembly and cargo loading of encapsulins in yeast. **a** Co-expression of encapsulin and targeted cargo in yeast results in self-assembly of cargo-loaded nanocompartments, with an internal diameter of 26 nm (structure from PDB: 4PT2). Encapsulin compartments can stabilize and co-localize cargo proteins, as well as allow for catalysis to occur within their interior. **b** Encapsulin *EncA* (Enc, 32.5 kDa) from *Myxococcus xanthus* can be purified to homogeneity from yeast, as determined by SDS-PAGE and TEM. Full gel image is shown in Supplementary Fig. 1. Scale bars: zoom out: 400 nm, zoom in: 50 nm. **c** Heterologous proteins such as mNeonGreen (mNeon, 28.4 kDa) can be packaged inside encapsulin compartments, as determined by SDS-PAGE and TEM on the co-purified sample. Full gel image is shown in Supplementary Fig. 2. M1 = Color Prestained Protein Ladder, Broad Range (11–245 kDa, NEB). Scale bars: zoom out: 200 nm, zoom in: 50 nm

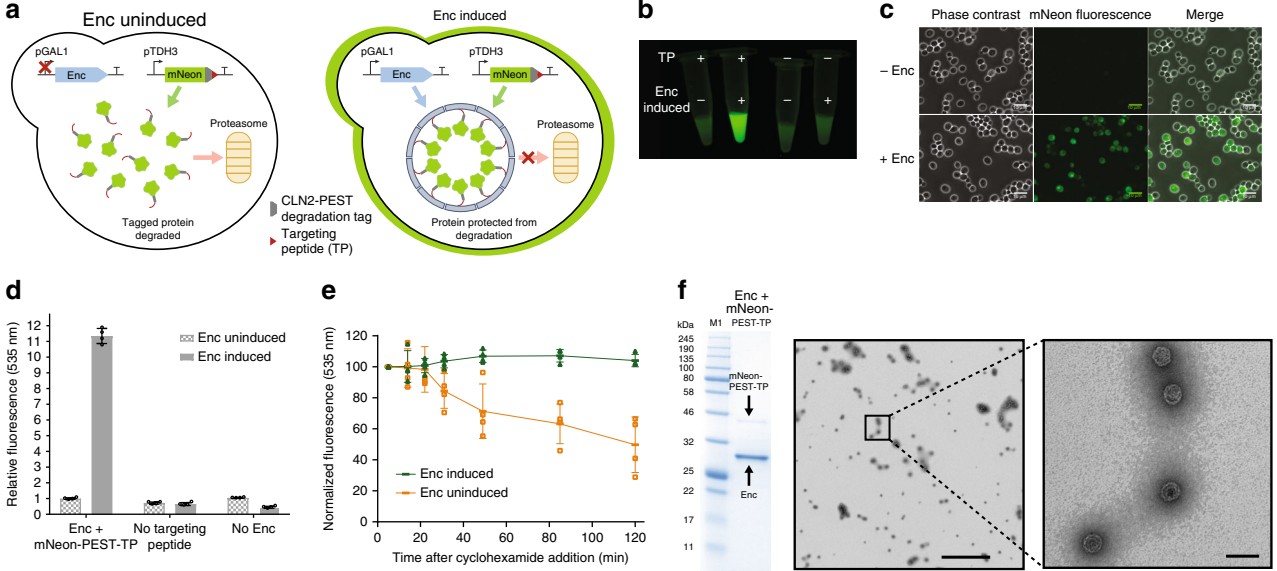

**Fig. 2** Protein stabilization in yeast using encapsulins. **a** Encapsulation of cargo proteins can enhance stability against proteolytic degradation. Destabilized mNeonGreen bearing a PEST degradation tag shows an increase in in vivo lifetime as determined using a variety of experimental techniques. **b** Bulk images of live yeast cells show greater fluorescence intensity corresponding to stabilized mNeonGreen cargo only when the targeting peptide (TP) and the encapsulin are present. **c** Images of live yeast cells by fluorescence microscopy show an elevated level of fluorescence only in the presence of encapsulin. Scale bar represents 10 μm. **d** Bulk plate-reader fluorescence intensity measurements of live yeast cells show an 11-fold increase in destabilized mNeonGreen cargo levels only when the TP and the encapsulin are present. **e** Upon inhibiting protein synthesis using cyclohexamide, destabilized mNeonGreen was protected from degradation only when encapsulated. **f** Isolation of the loaded encapsulins from yeast showed co-purification with destabilized mNeonGreen (mNeon-PEST-TP, 47.6 kDa) by SDS-PAGE and TEM. Full gel image is shown in Supplementary Fig. 2. M1 = Color Prestained Protein Ladder, Broad Range (11–245 kDa, NEB). Error bars represent the standard deviation of four biological replicates. Scale bars: zoom out: 800 nm, zoom in: 50 nm

from the cell lysate, followed by size-exclusion and ion-exchange chromatography, resulting in a pure sample of encapsulin as determined by SDS-PAGE (Fig. 1b and Supplementary Fig. 2). Under native PAGE conditions, the size of the encapsulin particle was > 1 MDa, consistent with the formation of a self-assembled capsid structure (Supplementary Fig. 5). Transmission electron microscopy (TEM) of negatively stained samples confirmed that the purified encapsulins were highly homogeneous spherical capsids with the expected diameter of 32 nm[14] (Fig. 1b and Supplementary Fig. 6).

In vivo self-assembly of protein cargo inside encapsulins was demonstrated using the fluorescent protein mNeonGreen with a C-terminal fused TP (sequence PEKRLTVGSLRR) under the control of the constitutive TDH3 promoter (Fig. 1c). After encapsulin induction and subsequent purification of the capsids from yeast, co-purification of mNeonGreen with encapsulin was observed by SDS-PAGE (Fig. 1c). The cargo loading percentage relative to encapsulin was estimated to be 24% (~43 molecules per compartment) based on gel densitometry. Fluorescence was also confirmed to be associated with the assembled capsids by in-gel fluorescence of the purified encapsulin band on a native PAGE gel (Supplementary Fig. 7). Confirmation that the cargo-loaded encapsulins had assembled as expected was obtained by TEM (Fig. 1c). Furthermore, the purified encapsulins were remarkably stable over time, with minimal deterioration observed by native PAGE and TEM despite storage for 2 months at 4 °C in Tris buffer (Supplementary Fig. 8).

**Protection of encapsulated cargo from degradation in vivo.** Cargo proteins packaged inside encapsulin compartments were protected against proteolytic degradation (Fig. 2). A destabilized cargo protein was created by appending mNeonGreen with a C-terminal CLN2-PEST degradation tag, followed by the TP for encapsulation. Yeast cells expressing this unstable fusion protein (mN-PEST-TP) only showed high levels of in vivo fluorescence when co-expressed with encapsulin, as determined in bulk measurements (Fig. 2b) and by fluorescence microscopy of individual cells (Fig. 2c). Minimal fluorescence intensity was observed when the TP was removed or the encapsulin was not induced. Based on bulk plate-reader fluorescence measurements (Fig. 2d), an 11-fold increase was observed for cargo protein levels as a result of encapsulation.

The stabilization effect of encapsulation was also observed after inhibition of new protein synthesis (Fig. 2e). Upon inhibition using cyclohexamide, yeast cells expressing only the destabilized cargo showed a gradual loss of fluorescence over 2 h. In comparison, yeast cells co-expressing the cargo and encapsulin maintained a constant level of fluorescence over 2 h period. To confirm the integrity of the encapsulin compartments, the loaded compartments were co-purified from yeast as before, displaying associated fluorescence by PAGE, and proper assembly by TEM (Fig. 2f and Supplementary Fig. 9).

**Co-localization of split proteins within encapsulins.** Multiple heterologous proteins can be co-localized inside encapsulin compartments (Fig. 3a). Using an established split-Venus system[27], an elevated level of fluorescence was only observed when both split components were targeted for encapsulation, and the encapsulin gene itself was present and induced (Fig. 3b–d). A 2.5-fold increase in fluorescence intensity was observed, consistent with the fluorescence response previously reported when the split components are co-localized[27] (Fig. 3c). Co-encapsulation of two distinct proteins did not disturb encapsulin assembly as indicated by the high molecular weight band on native PAGE (Fig. 3e) and the readily assembled particles observed using TEM (Fig. 3f). The split components were estimated to have a cargo loading

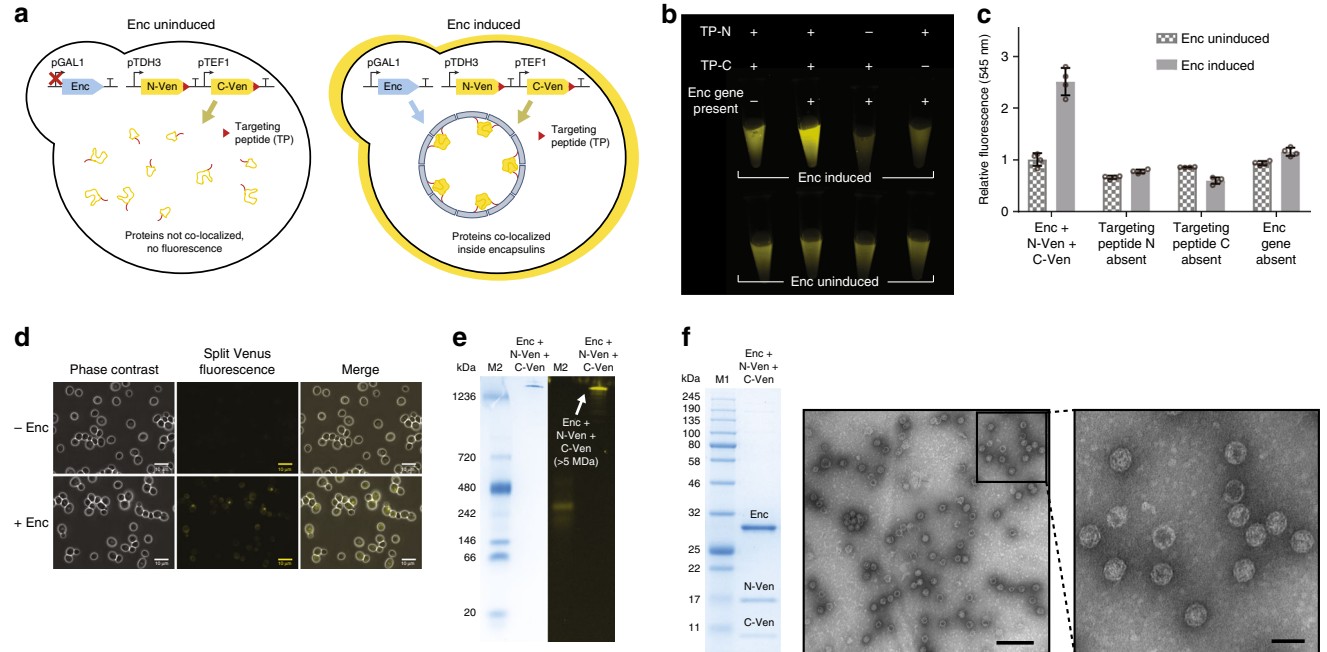

**Fig. 3** Protein co-localization and encapsulation in yeast. **a** Co-encapsulation of split-Venus components led to an increase in fluorescence intensity as determined by a variety of experimental techniques. **b** Bulk images of live yeast cells show greater fluorescence intensity only when both targeting peptides (TPs) are present and the encapsulin gene is present and induced. **c** Bulk plate-reader measurements indicate a 2.5-fold increase in fluorescence only with both TPs and encapsulin present. **d** Fluorescence microscopy of live yeast cells indicates greater fluorescence only in the presence of encapsulin. Scale bar represents 10 μm. **e** Native PAGE analysis with the left side showing a Coomassie-stained gel, and the right side showing fluorescence imaging of the same gel. The encapsulated split-Venus showed in-gel fluorescence associated with the high molecular weight assembled encapsulin nanocompartment band seen by Coomassie-staining (Enc + N-Ven + C-Ven). Full gel image is shown in Supplementary Fig. 10. **f** Purified encapsulins co-purified with the split-Venus proteins (N-Ven 19.4/C-Ven 11.6 kDa) by SDS-PAGE, and formed the expected compartments as imaged by TEM. Full gel image is shown in Supplementary Fig. 2. M2 = NativeMark Unstained Protein Standard (ThermoFisher). Error bars represent the standard deviation of four biological replicates. Scale bars: zoom out: 200 nm, zoom in: 50 nm

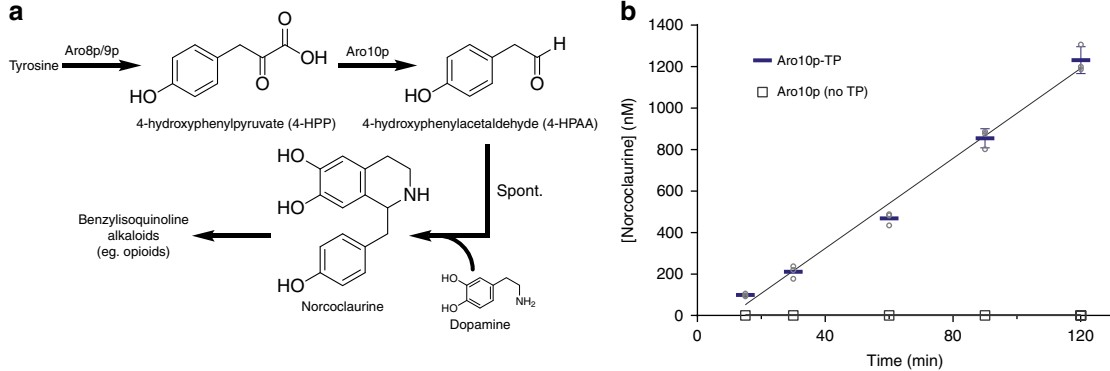

**Fig. 4** Enzymatic catalysis inside encapsulin compartments. **a** The Aro10p enzyme is involved in tyrosine catabolism, generating 4-hydroxyphenylacetaldehyde (4-HPAA) which can be measured by QTOF-LCMS after conversion to norcoclaurine by spontaneous reaction with dopamine. This pathway has previously been used to engineer the production of opioids in yeast[29]. **b** Purified encapsulin nanocompartments show enzymatic activity only when co-expressed with the version of Aro10p that is fused with the targeting peptide (TP). Encapsulins were purified from yeast strains co-expressing the encapsulin and either Aro10p-TP or non-targeted Aro10p (see Supplementary Methods and Supplementary Table 1 for strain construction). Error bars represent the standard deviation of three technical replicates

percentage of 42% for Ven-N and 30% for Ven-C (~76 and 54 per compartment, respectively) based on SDS-PAGE analysis (Fig. 3f).

**Catalytic turnover of an encapsulated yeast enzyme.** Finally, enzyme-loaded encapsulins were shown to be viable nanoreactors for catalytic processes. The assembled encapsulin structure[14]

contains small 5–10 Å sized pores, which in principle, can allow small molecule substrates and products to diffuse in and out of the compartment. The candidate enzyme chosen for encapsulation was Aro10p, a tetrameric pyruvate decarboxylase enzyme that is endogenous to yeast, participating in the catabolism of aromatic amino acids such as tyrosine (Fig. 4a). In particular, Aro10p catalyzes the decarboxylation of 4-hydroxyphenylpyruvate (4-HPP) to 4-hydroxyphenylacetaldehyde (4-HPAA)[28]. There is

great interest in the production of 4-HPAA in yeast, as its reaction with dopamine via Pictet-Spengler cyclization leads to norco-claurine, a key intermediate for the heterologous production of many valuable medicinal benzylisoquinoline alkaloids of the opioid family[29–31]. Two challenges associated with 4-HPAA production in yeast are instability due to endogenous aldehyde and alcohol dehydrogenases, and toxicity effects associated with aldehyde reactivity[32,33]. We sought to test if Aro10p could be encapsulated within encapsulins as a potential route toward addressing these challenges.

Encapsulin nanoreactors containing the Aro10p-TP enzyme displayed enzymatic 4-HPP decarboxylation activity (Fig. 4b). Comparing the encapsulins purified from yeast strains co-expressing either Aro10p-TP or Aro10p with no TP, only the encapsulins co-expressed with Aro10p-TP showed in vitro enzymatic activity in the presence of 4-HPP. Enzymatic activity of purified encapsulins was determined by spontaneous Pictet-Spengler cyclization of the 4-HPAA product with dopamine to give norcoclaurine, which could then be detected by QTOF-LCMS (Supplementary Fig. 11). To confirm the fidelity of the targeting process, the purified encapsulins with TP or no TP were compared by SDS-PAGE, indicating the presence of Aro10p only when the TP was present (Supplementary Fig. 12).

## Discussion

In this work, we establish encapsulins as a platform for engineering synthetic compartmentalization in yeast. There are several key features of encapsulins that distinguish them from other related compartmentalization systems currently being studied. The first is the ability to self-assemble with its associated cargo in vivo, using only a single repeating protein unit and short peptide tag. Other proteinaceous organelles, such as bacterial microcompartments, are comprised of many different protein subunits and thus entail a higher degree of complexity. Although significant progress has been made toward understanding the molecular principles governing these complex systems[1,10], our incomplete understanding is still a bottleneck for repurposing such systems as synthetic organelles.

The orthogonality of encapsulins in the context of eukaryotic organisms is another distinct aspect of our approach to synthetic compartmentalization. Recent reports have explored the localization of engineered proteins to eukaryotic compartments such as the peroxisome[25,26], mitochondria[23], and vacuoles[34]. In addition, the Arc protein of eukaryotic neurons was recently found to form virus-like compartments[35,36], which may have potential for future engineering applications. While large scale reprogramming may be tolerated for some native organelles, essential functions may be perturbed. The other restriction imposed when using native organelles is that the protein import mechanism, biochemical environment, and substrate permeability may also be difficult to modify. There are examples where the organelle environment is advantageous, such as using the oxidative environment of the mitochondria[23]. In a completely orthogonal synthetic compartment, these parameters can be tailored with much greater freedom.

Encapsulins present a tunable platform for maximizing the productivity of engineered pathways. The encapsulin targeting system enables co-localization of multiple enzymes. Co-localization could lead to significant rate enhancements for reactions involving unstable or toxic intermediates, or in situations where a high local concentration of the intermediate is required[37]. The levels of each enzyme within encapsulins could be controlled by modifying the TP sequence and hence the strength of its interaction with the encapsulin protein. Substrate accessibility may be tuned by engineering the residues adjacent to

the compartment pores. Furthermore, almost 1000 encapsulin variants have been reported[11], thereby providing compartments with different diameters, surface charges, pore sizes, and other biophysical properties that can be chosen.

In conclusion, we have shown that encapsulin compartments display many of the properties required for building synthetic organelles in eukaryotes. Encapsulin compartments can extend the lifetime of unstable proteins, and co-localize proteins to induce proximity effects. The encapsulin platform is also capable of serving as a nanoreactor, with encapsulated enzymes displaying catalytic activity. Taken together, the encapsulin system is modular and robust, with potential applications for enhancing protein production and metabolic engineering in yeast. This work now paves the way for future studies on controlling new enzymatic chemistry within encapsulins, and the integration of encapsulin organelles into engineered yeast metabolism.

## Methods

**Molecular biology and cloning**. All inserts were synthesized as codon-optimized gBlocks (IDT), and Sanger sequencing was performed by Genewiz or Eton Bioscience. All plasmids were cloned by Gibson assembly using NEBuilder HiFi DNA Assembly Master Mix (NEB). The destination vector for the inserts was the 2μ plasmid pAG423GAL-ccdB (Addgene #14149). See Supplementary Methods and Supplementary Table 1 for further details of the construction, and the associated GenBank files for full sequence details. Plasmids were first cloned in 5-alpha competent *E. coli* (NEB), isolated by miniprep, and then transformed into the CEN.PK2-1D strain of *S. cerevisiae* (Euroscarf) using the high efficiency LiAc/SS carrier DNA/PEG method described by Gietz and Schiestl[38]. Linear constructs were obtained from synthetic gBlocks and transformed directly into CEN.PK2-1D strain of *S. cerevisiae* (Euroscarf) using the high efficiency LiAc/SS carrier DNA/PEG method described by Gietz and Schiestl[38]. Cassettes were directed to disrupt the HO locus of the yeast genome. Selection for the KanMX resistance marker was carried out on G418 plates. Integrants were confirmed by PCR and sequencing of the integrated cassette, starting from regions outside the cassette.

**Encapsulin expression and purification**. Overnight 5 mL liquid cultures of yeast strains in synthetic defined dropout media were diluted into 50 mL of fresh media and grown at 30 °C for 18–24 h. Cells were resuspended in 6 mL PBS buffer and lysed using glass beads, and then sodium chloride and PEG-8000 were added to the soluble fraction to a final concentration of 0.5 M and 8%, respectively. After sitting for 15 min on ice, the precipitate was isolated, redissolved in 2 mL PBS buffer, and purified by size exclusion using a HiPrep 16/60 Sephacryl S-500 HR column (GE Healthcare) in PBS buffer (1 mL/min) on an AKTA Explorer (Amersham Biosciences). The encapsulin fractions were concentrated using Amicon Ultra-15 Centrifugal Filter Units with Ultracel-100 membrane (Millipore), then diluted in 2 mL of 20 mM Tris buffer at pH 8. Ion-exchange chromatography using a HiPrep DEAE FF 16/10 column (GE Healthcare) resulted in the fully purified encapsulin sample for ion-exchange. The gradient used for ion-exchange was as follows: 100% A for 0–100 mL, 100% A to 50% A + 50% B for 100–200 mL, 100% B for 200–300 mL, 100% A for 300–400 mL; where A is 20 mM Tris pH 8, B is 20 mM Tris pH 8 with 1 M NaCl (flow rate: 3 mL/min). Examples of purification chromatographs can be found in Supplementary Figs. 13 and 14.

**Polyacrylamide gel electrophoresis**. SDS-PAGE was run using Novex Wedge-Well 14% Tris-Glycine Mini Gels (Invitrogen), staining with Coomassie Brilliant Blue. Native PAGE was run using NativePAGE™ 3–12% Bis-Tris Protein Gels (Invitrogen), running either under regular Bis-Tris buffer conditions or using NativePAGE running buffers (Invitrogen) for Blue Native PAGE. Color Prestained Protein Standard, Broad Range 11–245 kDa (NEB) was used as a ladder for SDS-PAGE (marked as "M1"), while NativeMark Unstained Protein Standard (Life Technologies) was used for native PAGE (marked as "M2"). Gel images were captured on a ChemiDoc MP Imaging System (Bio-Rad), using the accompanying Image Lab software to approximate band intensities for densitometry measurements. Gel densitometry was carried out using the in-built quantification tools on the ImageLab software (Bio-Rad).

**Mass spectrometry**. Protein identification of SDS-PAGE gel bands was carried out at the HMS Taplin Mass Spectrometry Facility. Coomassie blue stained gels were destained. Single bands were excised with as little excess as possible. Excised gel bands were cut into ~1 mm³ pieces. Gel pieces were washed and dehydrated with acetonitrile for 10 min. Pieces were then completely dried in a speed-vac. Rehydration of the gel pieces was done with 50 mM $NH_4HCO_3$ solution containing 12.5 ng/μL modified sequencing-grade trypsin (Promega) at 4 °C. After 45 min, excess trypsin solution was removed and replaced with 50 mM $NH_4HCO_3$ solution to just cover the gel pieces. Samples were incubated at 37 °C overnight. Peptides

were extracted by removing the $NH_4HCO_3$ solution, followed by one wash with 50% acetonitrile and 1% formic acid. The extracts were then dried in a speed-vac (1 h). The samples were stored at 4 °C until analysis. On the day of analysis the samples were reconstituted in 5–10 µL of high-performance liquid chromatography (HPLC) solvent A (2.5% acetonitrile, 0.1% formic acid). A nanoscale reverse-phase HPLC capillary column was created by packing 2.6 µm C18 spherical silica beads into a fused-silica capillary (100 µm inner diameter, 30 cm length) with a flame-drawn tip. After equilibrating the column, each sample was loaded via a Famos auto sampler (LC Packings) onto the column. A gradient was formed and peptides were eluted with increasing concentrations of solvent B (97.5% acetonitrile, 0.1% formic acid). As peptides eluted they were subjected to electrospray ionization and then entered into an LTQ Orbitrap Velos Pro ion-trap mass spectrometer (Thermo Fisher Scientific). Peptides were detected, isolated, and fragmented to produce a tandem mass spectrum of specific fragment ions for each peptide. Peptide sequences (and hence protein identity) were determined by matching protein databases with the acquired fragmentation pattern using Sequest (Thermo Fisher Scientific). All databases include a reversed version of all the sequences, and the data were filtered to between a 1% and 2% peptide false discovery rate.

**Transmission electron microscopy**. Electron microscopy was conducted on a Tecnai G2 Spirit BioTWIN. Encapsulin samples were diluted to approximately 0.1 mg/mL for adsorption onto Formvar carbon coated gold grids 200 mesh FCF200-Au (EMS) after glow discharge. Excess sample was removed by blotting on filter paper (Whatman). Uranyl formate or uranyl acetate was then applied for negative staining. Particle size distributions for selected TEM images can be found in Supplementary Fig. 15.

**Fluorescence measurements**. Bulk yeast images—Cells were inoculated into 5 mL of SD-His and grown overnight. After normalizing for OD, the cells were resuspended in 2 mL of water, and 200 µL of this suspension was added to 2 mL of either SD-His or induction media. After 24 h growth at 30 °C, cells were pelleted and resuspended in water normalizing to an OD value of 28. Images were taken on a ChemiDoc MP Imaging System (Bio-Rad).

Plate-reader measurements—Samples were prepared in the same manner as for bulk imaging, normalizing to an OD value of 1.5, with fluorescence intensity measurements carried out on a Synergy Neo plate reader (BioTek), with excitation/emission wavelengths of 500/535 nm for mNeonGreen and 515/545 nm for Venus. Measurements were carried out in independent biological quadruplicate experiments, each consisting of technical triplicates.

Fluorescence microscopy—Cells prepared for bulk imaging were also imaged directly on a Nikon TE 2000 microscope in glass bottom dishes (MatTek, 35 mm, uncoated, no. 1.5) under agar pad. Microscope light source power, detector gain, and image processing settings were kept consistent between images and samples to ensure the validity of any comparative conclusions drawn. Extra images at lower magnification can be found in Supplementary Figs. 16 and 17.

Cyclohexamide chase experiment—Cells were first grown using the same protocol as for bulk imaging described above. From the 24 h induced and non-induced cultures, 1 mL of each culture was pelleted and resuspended into 10 mL of fresh media (OD ~0.8) and grown at 30 °C for 40 min. After this, 100 µg/µL cyclohexamide was added to each culture, and at each time point, 400 µL aliquots were taken, resuspended in 100 µL $H_2O$ and snap frozen for later measurement. Fluorescence intensity measurements were obtained by the plate-reader method described above. Measurements were carried out in independent biological quadruplicate experiments, each consisting of technical triplicates.

**Enzymatic assays**. Enzyme assays were conducted in technical triplicate. The assay conditions were 100 mM potassium phosphate buffer pH 7, 1 mM $MgCl_2$, 0.5 mM thiamine pyrophosphate, 1 mM dopamine hydrochloride, 1 mM 4-HPP and 75 µg/mL encapsulin. Reactions were conducted at 30 °C in 1 mL volumes. At each time point, 100 µL was removed from the reaction and 100 µL of acetonitrile (MeCN) was added. Any precipitated debris was pelleted, and then 5 µL of the supernatant was added to 45 µL of water to give the final sample ready for QTOF-LCMS analysis.

Norcoclaurine production was measured on a QTOF-LCMS (Agilent 6530), running samples on a Orpak CDBS 453 column (Shodex). The method used for analysis was: 0–9 min 0% B, 9–11 min 0 to 95% B, 11–14 min 95% B, 14–16 min 95 to 0% B, 16–23 min 0% B (flow rate 0.5 mL/min; A = 95% $H_2O$ + 5% MeCN with 0.1% formic acid, B = 100% MeCN). The MS acquisition parameters were as follows: positive ion mode, gas temperature: 325 °C, drying gas: 10 L/min, nebulizer: 12 psig, VCap: 3500 V, mass range: 100–1000 $m/z$, acquisition rate: 2 spectra/s, acquisition time: 500 ms/spectrum. Norcoclaurine standards for generating a standard curve were obtained from Toronto Research Chemicals (Supplementary Fig. 18).

**Data availability**. The sequence data, in GenBank format, for all the constructs created in this study are available in Supplementary Data 1. All other data that support the findings of this study are available from the corresponding author upon request.

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

## Acknowledgements

Y.H.L. acknowledges funding from the Wellcome Trust (107402/Z/15/Z). T.W.G. was supported by a Leopoldina Research Fellowship (LPDS 2014-05) from the German National Academy of Sciences. This work was supported by the National Science Foundation (NSF) Synthetic Biology of Yeast grant (MCB-1330914). T.W.G and P.A.S. acknowledge support from the Wyss Institute for Biologically Inspired Engineering at Harvard.

## Author contributions

Y.H.L., T.W.G., and P.A.S. designed the research and wrote the manuscript. Y.H.L., T.W.G., and W.J.A. performed the experiments.

## Additional information

**Competing interests:** The authors declare no competing interests.

