## [Peer Review File · Nature Communications]

Reviewers' comments:

Reviewer #1 (Remarks to the Author):

This is a simply written manuscript describing a well conceived and timely use of encapsulin nano compartments as synthetic organelles in eukaryotes. Overall the manuscript presents a solid foundational step for the use of encapsulin nano compartments in yeast and the data presented represents well-executed experiments that support the authors conclusions.

However, the manuscript requires significant editing and revision before publication to strengthen the arguments made and to allow the reader to fully interpret the data presented.

I recommend the following changes before publication:

General:

The figure legends are not sufficiently detailed for the reader to fully understand the data presented; particularly in the supplementary data, where the legends are entirely absent. Please revise the legends of all figures to be more descriptive and allow the reader to appreciate the results presented.

Gels across the manuscript are inconsistently labelled with lanes and molecular weight markers. Please tidy these up for consistency. Please also label bands of interest for the reader.

Introduction:

1. Given the recent publications on the virus-like capsid present in human neurones and its role in signalling/transfer of nucleic acid. Is it reasonable to say that encapsulin nano compartments are 'completely' orthogonal to endogenous eukaryotic compartments? Do these eukaryotic virus-like shells share any sequence identity with encapsulin nano compartments? If not it is still worth mentioning these for context.

Results:

Self-assembly...

Did the authors perform any mass-spectrometry experiments to determine the accurate masses of the purified encapsulins (c.f. gels in S3.3). For any assertion that the monomers are produced correctly, LC-MS analysis is necessary to rule out any mis-translation or PTM.

Analysis of capsid size presented in fig 1B and S4.1 would be well supported with some measurements and a size-distribution histogram calculated from these.

The cargo loading percentage as described represents bulk measurements and are subject to the standard kind of errors common to all gel-based densitometry studies. Are there micrographs available for the authors to calculate what % of encapsulins appear to have cargoes? This would be useful and interesting data.

Where are the encapsulins purified from in the data presented in Figure 4? Or was this data collected on in vivo measurements?

Methods

What were the final destination vectors for the inserts? Please clarify this in the methods.

Fluorescence measurements: What final OD were the cells normalised to before the fluorescence measurements were taken?

Enzymatic assays:

Were all measurements really taken as biological triplicates, or as technical replicates?

Define MeCN for the reader.

Supplementary data:

Please give proper figure legends for all supplementary items.

Cloned plasmids: Please give more information on the 2u plasmid? What is it? What's the source of this plasmid?

Figure S3.2 is rather poorly presented. Please clean up these graphs and add the salt-gradient curve to the FPLC trace for the ion exchange.

Yeast fluorescence measurements: The protocol is entirely missing. Please insert this into section 6 of the SI.

Catalytic nano reactor measurements: The HPLC traces are too small to interpret, please increase the size and label areas of interest for the reader's benefit.

Reviewer #2 (Remarks to the Author):

This manuscript describes the use of the *Myxococcus xanthus* encapsulin in *Saccharomyces cerevisiae*. The authors have shown that the encapsulin, produced via a single protein EncA, can be expressed and stably encapsulate the fluorescent protein, mNeonGreen, by TEM and through purification of the encapsulin and visualising the protein on a SDS-PAGE. The stability of the encapsulin and its' cargo (mNeonGreen) from protease treatment and storage was also shown. To test multiple heterologous protein co-localization, split-Venus system (N-Van and C-Van) was encapsulated. To further test whether the encapsulin can be used as a nanoreactor, a tagged tetrameric pyruvate decarboxylase, Aro10-TP, was encapsulated. The tetrameric pyruvate decarboxylase produces 4-hydroxyphenylacetaldehyde (4-HPAA) from 4-hydroxyphenylpyruvate which is then by spontaneous Pictet-Spengler cyclization of the 4-HPAA with dopamine produces norcoclaurine, which is an important intermediate for other alkaloid compounds, e.g. opioids. The authors show a successful stable expression and activity of Aro10-TP and can detect norcoclaurine produced only when it is encapsulated.

The manuscript shows work using bacterial encapsulin as an alternative compartmentalisation strategy that can be used in yeast. However, there are several recent publications on encapsulins and their use as a nanoreactor. It would strengthen this manuscript, if the authors could show more conclusively, the novelty of this system in yeast.

Some comments:

- Line 71: "the identity of the SDS-PAGE band was confirmed by MS". Please show the data.
- Line 85, Figure 1 C: Please add fluorescent TEM of the purified mNeon encapsulated encapsulins (as shown in Figure 2C) to show successful encapsulation and purification.
- Figure 2E: With the current colouring of the induced and uninduced cells, it is very difficult to differentiate them. Please use contrasting colours.
- Figure 2F: Please show the mNeon fluorescent in the TEM.
- Figure 3A: The co-localization of the N-Van and C-Van in the encapsulin is shown as the two tags on both ends of the multienzyme complex. How sure is this structure? Please expand on this, using known enzyme complex of N-Van and C-Van and the relation of the tags in the crystal structure. If it is not sure, then please do not use a figure like this as it tends to be misunderstood.
- The authors comment that "4-HPAA production in yeast are instability due to endogenous aldehyde and alcohol dehydrogenases, and toxicity effects associated with aldehyde reactivity". If

this is the case, then the growth of the yeast harbouring only the Aro10 could be different from that of Aro10-TP+encapsulin. Also the appropriate comparison would be Aro10-TP without the encapsulin.

Please add either a growth curve of the yeast + Aro10-TP with and without encapsulin. Or change the Figure 4B to show norcochlorine production/cell.

- Supplementary figure 3.8: make the arrow larger so it is visible. Also annotate all the bands that are present on the gel (Aro-TP, Env, etc).
- In Supplementary figure 3.8, it is written "Aro10-TP is only present as a single copy" why is this not on a plasmid? And why are the previous cargos (mNeon and split-Venus system) on a plasmid? Please also use plasmid for the ARO10-TP and single copy for mNeon and split-Venus system to be consistent.
- Please explain all the additional bands seen in all the SDS-PAGE shown in the supplementary file.
- Amend "ARO10" in the figures and supplementary files to "Aro10" to be consistent with the main text.
- Supplementary figure S7.2 left: the overlay of the QTOF-LCMS is not legible. Please amend the figure so that one can see both standard and sample chromatograms.
- Line 186. Please expand on the "thousands of encapsulin" and also cite a reference.
- There are several interesting manuscripts on the use of encapsulin that are not cited:
 - i. Structural Characterization of Native and Modified Encapsulins as Nanoplatfoms for in Vitro Catalysis and Cellular Uptake. Putri RM, Allende-Ballester C, Luque D, Klem R, Rousou KA, Liu A, Traulsen CH, Rurup WF, Koay MST, Castón JR, Cornelissen JJLM. ACS Nano. 2017 Dec 26;11(12):12796-12804. doi: 10.1021/acsnano.7b07669. Epub 2017 Dec 1.
 - ii. Successful PEGylation of hollow encapsulin nanoparticles from *Rhodococcus erythropolis* N771 without affecting their disassembly and reassembly properties. Sonotaki S, Takami T, Noguchi K, Odaka M, Yohda M, Murakami Y. Biomater Sci. 2017 May 30;5(6):1082-1089. doi: 10.1039/c7bm00207f.
 - iii. Assembly and Mechanical Properties of the Cargo-Free and Cargo-Loaded Bacterial Nanocompartment Encapsulin. Snijder J, Kononova O, Barbu IM, Uetrecht C, Rurup WF, Burnley RJ, Koay MS, Cornelissen JJ, Roos WH, Barsegov V, Wuite GJ, Heck AJ. Biomacromolecules. 2016 Aug 8;17(8):2522-9. doi: 10.1021/acs.biomac.6b00469. Epub 2016 Jul 7.
 - iv. Recombinant expression and purification of "virus-like" bacterial encapsulin protein cages. Rurup WF, Cornelissen JJ, Koay MS. Methods Mol Biol. 2015;1252:61-7. doi: 10.1007/978-1-4939-2131-7_6. PMID: 25358773

Response to reviewer comments

Reviewer #1 (Remarks to the Author):

This is a simply written manuscript describing a well conceived and timely use of encapsulin nano compartments as synthetic organelles in eukaryotes. Overall the manuscript presents a solid foundational step for the use of encapsulin nano compartments in yeast and the data presented represents well-executed experiments that support the authors conclusions.

However, the manuscript requires significant editing and revision before publication to strengthen the arguments made and to allow the reader to fully interpret the data presented.

I recommend the following changes before publication:

General:

The figure legends are not sufficiently detailed for the reader to fully understand the data presented; particularly in the supplementary data, where the legends are entirely absent. Please revise the legends of all figures to be more descriptive and allow the reader to appreciate the results presented.

A significant amount of explanatory text is now added to all the figures, especially the supplementary. Please see the red text in the manuscript files.

Gels across the manuscript are inconsistently labelled with lanes and molecular weight markers. Please tidy these up for consistency. Please also label bands of interest for the reader.

The labels for the lanes and bands in the gels are now fully consistent (eg. using mNeon always instead of mN, N-Ven + C-Ven rather than spVenus). To further increase clarity, the labels used on the PAGE gels are now also written in the figure captions of Figs 1,2,3.

The only gel images in the entire manuscript that previously did not have any labels are Supplementary Figures 1 and 10 (Figs S3.9/3.10 in old version of SI), which are the fully uncropped gels just to prove that we did not unfairly crop any images. We have now added labels to these gels.

Every figure already also has the ladder lanes marked, and the weights of every ladder band marked. Just to be doubly clear, the name of the SDS-PAGE ladder for M1 and the Native-PAGE ladder M2 is now included in the Figure captions in the main text and SI.

Text added: "M1 = Color Prestained Protein Ladder, Broad Range (11-245 kDa, NEB)." and "M2 = NativeMark Unstained Protein Standard (ThermoFisher)."

Introduction:

1. Given the recent publications on the virus-like capsid present in human neurones and its role in signalling/transfer of nucleic acid. Is it reasonable to say that encapsulin nano compartments are 'completely' orthogonal to endogenous eukaryotic compartments? Do these eukaryotic virus-like shells share any sequence identity with encapsulin nano compartments? If not it is still worth mentioning these for context.

Encapsulins and the eukaryotic capsids share no sequence homology and possess different folds, their only commonality is that they are both being capsid-forming proteins.

On the structural level, these recently reported eukaryotic Arc capsids are homologous to the gag polyprotein, which is the fold found in retroviruses such as HIV (which is somewhat irregularly shaped in terms of symmetry). This fold structure is different from the encapsulin, which has the bacteriophage HK97 fold, which is a highly regular and symmetrical fold.

At the sequence level, encapsulins and arc/gag proteins share no measurable similarity. A phylogenetic tree (neighbor-joining, outgroup: HK97 bacteriophage) based on a ClustalOmega alignment of fly and mouse arc proteins and the M. xanthus encapsulin is shown here.

Nevertheless, we agree that this work has relevance, and we now include text and references to this work to add context as suggested by the Reviewer.

Text added to discussion section paragraph 2: "In addition, the Arc protein of eukaryotic neurons was recently found to form virus-like compartments^{31,32}, which may have potential for future engineering applications"

Results:

Self-assembly...

Did the authors perform any mass-spectrometry experiments to determine the accurate masses of the purified encapsulins (c.f. gels in S3.3). For any assertion that the monomers are produced correctly, LC-MS analysis is necessary to rule out any mis-translation or PTM.

We performed LC/MS/MS analysis of the encapsulin bands extracted from SDS-PAGE gels, as previously stated in SI Section 1 (old version of SI). For clarity, we have now moved this information to its own dedicated Supplementary Figure 4 of the SI document, along with the corresponding summary of the data as requested by Reviewer 2.

Analysis of capsid size presented in fig 1B and S4.1 would be well supported with some measurements and a size-distribution histogram calculated from these.

Size distribution histograms have now been made for Fig 1B and S4.1. These are now presented in Supplementary Figure 15.

The cargo loading percentage as described represents bulk measurements and are subject to the standard kind of errors common to all gel-based densitometry studies. Are there micrographs available for the authors to calculate what % of encapsulins appear to have cargoes? This would be useful and interesting data.

We agree that gel-based densitometry studies are subject to some degree of error and the numbers should be taken as estimates (and this is the reason we use the word 'estimate' in the main text).

We do also have more micrographs from negative stain TEM. However, such calculations from TEM micrographs are extremely inaccurate as the appearance of the encapsulin interior can vary greatly depending on the nature of the negative-staining process, and we do not wish to over-interpret any artifacts arising from this. We believe that gel densitometry is thus the better estimate for cargo loading.

Where are the encapsulins purified from in the data presented in Figure 4? Or was this data collected on in vivo measurements?

The encapsulins were purified from yeast strains co-expressing the encapsulin and Aro10p either with or without the targeting peptide. This is now clarified in the figure caption.

Text added to Figure 4 caption: "Encapsulins were purified from yeast strains co-expressing the encapsulin and either Aro10p-TP or non-targeted Aro10p (see Supplementary Methods and Supplementary Table 1 for strain construction)."

Methods

What were the final destination vectors for the inserts? Please clarify this in the methods.

The final destination vector for the inserts was mentioned in the Supporting Information Section 2 (old version of SI). It is now also included in the main text Methods for clarity.

Text added to Methods section under the 'cloning' heading: "The destination vector for the inserts was the 2 μ plasmid pAG423GAL-ccdB (Addgene #14149). See Supplementary Methods and Supplementary Table 1 for further details of the construction, and the associated GenBank files for full sequence details."

Fluorescence measurements: What final OD were the cells normalised to before the fluorescence measurements were taken?

For the plate reader measurements, the cells were normalized to an OD value of 1.5. For the bulk

measurements, the cells were normalized to an OD value of 28 due to the lower sensitivity of the ChemiDoc imaging system.

Text added to Methods section under the 'fluorescence measurements' heading: "normalizing to an OD value of 28" and "normalizing to an OD value of 1.5".

Enzymatic assays:

Were all measurements really taken as biological triplicates, or as technical replicates?

Define MeCN for the reader.

These were indeed technical replicates, and not biological triplicates as they were mislabeled initially.

The text is now corrected. Also, MeCN is now changed to acetonitrile (MeCN) in the text.

Text added to Methods section under the 'enzymatic assays' heading: "in technical triplicate" and "acetonitrile (MeCN)"

Supplementary data:

Please give proper figure legends for all supplementary items.

As mentioned before, a significant amount of explanatory text is now added to the supplementary figures. Please see the red text in the manuscript files.

Cloned plasmids: Please give more information on the 2u plasmid? What is it? What's the source of this plasmid?

A full description of the plasmid, including its entire sequence, the method of cloning, and the backbone it was derived from (and Addgene source), are given in the Supplementary Methods, Supplementary Table 1 and accompanying zip file.

Figure S3.2 is rather poorly presented. Please clean up these graphs and add the salt-gradient curve to the FPLC trace for the ion exchange.

The text has been enlarged and the peaks labelled for Supplementary Figures 13/14 (previously Figure S3.2 in old version of SI).

The salt-gradient that was already mentioned in the Methods section (under Encapsulin expression and purification) is now also plotted in graphical form.

Yeast fluorescence measurements: The protocol is entirely missing. Please insert this into section 6 of the SI.

The full protocols were actually moved from the SI into the main text methods section (see "fluorescence measurements" section of the main text Methods). We had mistakenly left the 'protocol' heading in section 6 SI (old version of SI), hence the confusion. We have now removed this erroneous heading, but keep the extra images in Supplementary Figures 16/17, which just show zoomed out versions of the images shown in the main text.

Catalytic nano reactor measurements: The HPLC traces are too small to interpret, please increase the size and label areas of interest for the reader's benefit.

The LCMS traces in Supplementary Figure 11 have now been updated with a zoom-in of the critical section, labelled with the two norcoclaurine peaks (one for each enantiomer).

Reviewer #2 (Remarks to the Author):

This manuscript describes the use of the *Myxococcus xanthus* encapsulin in *Saccharomyces cerevisiae*. The authors have shown that the encapsulin, produced via a single protein EncA, can be expressed and stably encapsulate the fluorescent protein, mNeonGreen, by TEM and through purification of the encapsulin and visualising the protein on a SDS-PAGE. The stability of the encapsulin and its' cargo (mNeonGreen) from protease treatment and storage was also shown. To test multiple heterologous protein co-localization, split-Venus system (N-Van and C-Van) was encapsulated. To further test whether the encapsulin can be used as a nanoreactor, a tagged tetrameric pyruvate decarboxylase, Aro10-TP, was encapsulated. The tetrameric pyruvate decarboxylase produces 4-hydroxyphenylacetaldehyde (4-HPAA) from 4-hydroxyphenylpyruvate which is then by spontaneous Pictet-Spengler cyclization of the 4-HPAA with dopamine produces norcoclaurine, which is an important intermediate for other alkaloid compounds, e.g. opioids. The authors show a successful stable expression and activity of Aro10-TP and can detect norcoclaurine produced only when it is encapsulated.

The manuscript shows work using bacterial encapsulin as an alternative compartmentalisation strategy that can be used in yeast. However, there are several recent publications on encapsulins and their use as a nanoreactor. It would strengthen this manuscript, if the authors could show more conclusively, the novelty of this system in yeast.

Previous work on encapsulins as nanoreactors uses the native enzymatic cargo of the encapsulin (*B. linens* DyP enzyme is the native cargo of that encapsulin - 10.1021/acsnano.7b07669). Here we are using engineered enzyme cargo that has interesting applications in metabolic engineering. Thus, our study paves the way for future optimization of important engineered pathways in yeast.

Some comments:

- Line 71: "the identity of the SDS-PAGE band was confirmed by MS". Please show the data.

We have now added the summary data for the LC/MS/MS analysis of the encapsulin band extracted from SDS-PAGE in a dedicated Supplementary Figure 4 (also now referenced in the text). We have not included all the exhaustive raw data as this would consist of hundreds of MS fragmentation plots such as the one below (the example given corresponds to one MS hit for the N-terminal region of the protein):

Sequence
M⁺ PLEPRFM⁺ PDLGHAENFLR

Predicted Fragmentation Pattern

+1					
Seq #	b: Δ Error	b	y	y: Δ Error	+1
M ⁺ 1	---	148,043	---	---	20
P 2	82,253	245,095	2233,086	---	19
L 3	-212,876	358,180	2136,033	---	18
E 4	-92,061	487,222	2022,949	---	17
P 5	-1164,374	584,275	1893,906	---	16
H 6	175,669	721,334	1796,854	---	15
F 7	-1089,499	868,402	1659,795	---	14
M ⁺ 8	-0,625	1015,438	1512,726	105,018	13
P 9	457,798	1112,490	1365,691	249,455	12
D 10	84,058	1227,517	1268,638	-657,069	11
F 11	-112,538	1374,586	1153,611	119,921	10
L 12	192,730	1487,670	1006,543	239,275	9
G 13	152,687	1544,691	893,459	57,665	8
H 14	---	1681,750	836,437	303,891	7
A 15	---	1752,787	699,378	-10,642	6
E 16	---	1881,830	628,341	157,641	5
N 17	---	1995,873	499,299	-50,736	4
P 18	---	2092,926	385,256	553,286	3
L 19	---	2206,010	288,203	---	2
R 20	---	---	175,119	---	1

+2					
Seq #	b: Δ Error	b	y	y: Δ Error	+1
M ⁺ 1	---	74,525	---	---	20
P 2	---	123,051	1117,047	707,348	19
L 3	---	179,593	1068,520	368,585	18
E 4	---	244,115	1011,978	414,582	17
P 5	---	292,641	947,457	509,730	16
H 6	151,417	361,171	898,930	336,786	15
F 7	---	434,705	830,401	405,596	14
M ⁺ 8	-38,228	508,222	756,867	956,299	13
P 9	---	556,749	683,349	290,727	12
D 10	411,467	614,262	634,823	-1247,291	11
F 11	461,351	687,796	577,309	384,568	10
L 12	1271,226	744,339	503,775	220,616	9
G 13	538,694	772,849	447,233	1296,308	8
H 14	363,100	841,379	418,722	-148,657	7
A 15	814,683	876,897	350,193	150,289	6
E 16	704,904	941,419	314,674	---	5
N 17	424,590	998,440	250,153	-3583,209	4
P 18	859,513	1046,966	193,132	---	3
L 19	305,644	1103,508	144,605	---	2
R 20	---	---	88,063	---	1

- Line 85, Figure 1 C: Please add fluorescent TEM of the purified mNeon encapsulated encapsulins (as shown in Figure 2C) to show successful encapsulation and purification.

The reviewer may have misunderstood what information can be obtained by TEM, and what Figure 2C represents. TEM cannot give any information about fluorescence, and Figure 2C does not show “fluorescent TEM”. Figure 2C is fluorescence microscopy of live yeast cells.

We have indeed shown that purified encapsulins containing mNeon are fluorescent. This data is in Supplementary Figure 7 (previously Figure S3.6 in old version of SI) where in-gel fluorescence is observed for the encapsulins in a Native-PAGE gel.

- Figure 2E: With the current colouring of the induced and uninduced cells, it is very difficult to differentiate them. Please use contrasting colours.

The colors are now changed from green-blue to green-orange for better contrast.

- Figure 2F: Please show the mNeon fluorescent in the TEM.

As stated before, TEM as a technique gives no information about fluorescence, so it is not appropriate to show the images with fluorescence.

- Figure 3A: The co-localization of the N-Van and C-Van in the encapsulin is shown as the two tags on both ends of the multienzyme complex. How sure is this structure? Please expand on this, using known enzyme complex of N-Van and C-Van and the relation of the tags in the crystal structure.

If it is not sure, then please do not use a figure like this as it tends to be misunderstood.

We believe the drawing in figure 3A does accurately reflect the structure of the split protein based on the following. Both C-termini of the two split components facing in the same direction of the beta barrel, based on the crystal structure of the fluorescent protein barrel itself. See Kodama and Hu

(Biotechniques 2010) from which this system was based on, and the Figure 1 in that paper (copied below) which shows the two termini pointing downwards – strand 7 and 11.

- The authors comment that “4-HPAA production in yeast are instability due to endogenous aldehyde and alcohol dehydrogenases, and toxicity effects associated with aldehyde reactivity”. If this is the case, then the growth of the yeast harbouring only the Aro10 could be different from that of Aro10-TP+encapsulin. Also the appropriate comparison would be Aro10-TP without the encapsulin. Please add either a growth curve of the yeast + Aro10-TP with and without encapsulin. Or change the Figure 4B to show norcochlorine production/cell.

This experiment was conducted *in vitro* on purified encapsulins, as stated in the final paragraph of the Results section. We are measuring the enzymatic activity of encapsulins filled with targeted Aro10p, and as a control to ensure we are not getting any background activity, comparing this to encapsulin that doesn't have targeted Aro10. To clarify that the enzymatic assays were done on purified encapsulins, we now also state this in the figure caption to complement the main body text.

Text added to Figure 4 caption: “B) Purified encapsulin nanocompartments...”

- Supplementary figure 3.8: make the arrow larger so it is visible. Also annotate all the bands that are present on the gel (Aro-TP, Env, etc).

The arrow is now larger, and the Aro10p bands and the Enc band are now labelled. Please note in the latest SI, this is now Supplementary Figure 12.

- In Supplementary figure 3.8, it is written “Aro10-TP is only present as a single copy” why is this not on a plasmid? And why are the previous cargos (mNeon and split-Venus system) on a plasmid? Please also use plasmid for the ARO10-TP and single copy for mNeon and split-Venus system to be consistent. For metabolic engineering applications, we felt that having the enzymes integrated as a single copy was preferred in terms of both stability of the construct, consistency of the expression level, and metabolic burden.

The previous cargos (mNeon and split Venus) were actually made on both a plasmid system and a single copy system. This is stated in the list of strains (Supplementary Table 1). The single copy integrated versions were used for all the functional experiments (stabilization, co-localization). The only use of the constructs with cargo on the high-copy plasmid was to get sufficient expression levels to show clear bands by SDS-PAGE to demonstrate co-purification. Thus, all activity assays were done with integrated genes.

- Please explain all the additional bands seen in all the SDS-PAGE shown in the supplementary file. These minor additional bands in Supplementary Figure 12 (previously Figure S3.8 in the old version of the SI) are due to the sample being purified only by size-exclusion (one-step), and not subsequently by ion-exchange (two-step), as indicated in the caption. See Supplementary Figure 2 for the comparison between the one-step and the two-step – in this gel, you can see the same additional bands.

These additional bands are minor impurities that co-elute at a similar size range, but do not affect functionality or conclusions of the experiment, as the control NoTP sample also contains these minor bands, but does not have enzymatic activity.

Text added to Supplementary Figure 12 caption: “so minor impurity bands remain in both samples, although these do not have any enzymatic activity.”

- Amend “ARO10” in the figures and supplementary files to “Aro10” to be consistent with the main text. We have amended Figure 4 and the supplementary. We now consistently use Aro10p for the protein, and ARO10 for the gene.

- Supplementary figure S7.2 left: the overlay of the QTOF-LCMS is not legible. Please amend the figure so that one can see both standard and sample chromatograms.

This figure is now called Supplementary Figure 18 in the new SI version. It is now amended to also include the separated chromatograms corresponding to the overlay.

- Line 186. Please expand on the “thousands of encapsulin” and also cite a reference. We have identified now over a thousand encapsulins, but these are not yet in the literature. Therefore, we have amended this statement to reflect the current state of literature (ie. Almost one thousand), and included the reference to the paper on discovery of this range of encapsulins.

Text amended: “Furthermore, almost one thousand encapsulin variants have been reported¹¹, thereby...”

- There are several interesting manuscripts on the use of encapsulin that are not cited:
 - i. Structural Characterization of Native and Modified Encapsulins as Nanoplatforms for in Vitro Catalysis and Cellular Uptake. Putri RM, Allende-Ballester C, Luque D, Klem R, Rousou KA, Liu A, Traulsen CH, Rurup WF, Koay MST, Castón JR, Cornelissen JLM. ACS Nano. 2017 Dec 26;11(12):12796-12804. doi: 10.1021/acsnano.7b07669. Epub 2017 Dec 1.
 - ii. Successful PEGylation of hollow encapsulin nanoparticles from Rhodococcus erythropolis N771 without affecting their disassembly and reassembly properties. Sonotaki S, Takami T, Noguchi K, Odaka M, Yohda M, Murakami Y. Biomater Sci. 2017 May 30;5(6):1082-1089. doi: 10.1039/c7bm00207f.
 - iii. Assembly and Mechanical Properties of the Cargo-Free and Cargo-Loaded Bacterial Nanocompartment Encapsulin. Snijder J, Kononova O, Barbu IM, Uetrecht C, Rurup WF, Burnley RJ, Koay

MS, Cornelissen JJ, Roos WH, Barsegov V, Wuite GJ, Heck AJ. Biomacromolecules. 2016 Aug 8;17(8):2522-9. doi: 10.1021/acs.biomac.6b00469. Epub 2016 Jul 7.
iv. Recombinant expression and purification of "virus-like" bacterial encapsulin protein cages. Rurup WF, Cornelissen JJ, Koay MS. Methods Mol Biol. 2015;1252:61-7. doi: 10.1007/978-1-4939-2131-7_6. PMID: 25358773

These references have now been added to the introduction.

Text added to introduction paragraph 2: "Encapsulins have been the subject of several recent engineering and characterization studies¹⁷⁻²⁰,..."

REVIEWERS' COMMENTS:

Reviewer #1 (Remarks to the Author):

We thanks the authors for their thorough consideration of all the review comments and recommend publication without delay.

Reviewer #2 (Remarks to the Author):

The authors have addressed all the comments from the reviewers adequately and added substantial additional information.

I think the manuscript is now ready to be accepted.